# An abdominal spacer that does not require surgical removal and allows drainage of abdominal fluids in patients undergoing carbon ion radiotherapy

**Norio Kubo**[1]*, **Takehiko Yokobori**[2,3,4], **Ryo Takahashi**[4], **Hiroomi Ogawa**[4], **Navchaa Gombodorj**[2], **Naoya Ohta**[5], **Tatsuya Ohno**[6], **Hiroshi Saeki**[4], **Ken Shirabe**[1], **Takayuki Asao**[7]

1 Department of Hepatobiliary and Pancreatic Surgery, Graduate School of Medicine, Gunma University, Maebashi, Gunma, Japan, 2 Division of Integrated Oncology Research, Gunma University Initiative for Advanced Research (GIAR), Maebashi, Gunma, Japan, 3 Department of Innovative Cancer Immunotherapy, Gunma University, Maebashi, Gunma, Japan, 4 Department of General Surgical Science, Graduate School of Medicine, Gunma University, Maebashi, Gunma, Japan, 5 Division of Electronics and Informatics, Gunma University Graduate School of Engineering, Kiryu, Gunma, Japan, 6 Gunma University Heavy Ion Medical Center, Maebashi, Gunma, Japan, 7 Big Data Center for Integrative Analysis, Gunma University Initiative for Advance Research, Maebashi, Gunma, Japan

* nkubo@gunma-u.ac.jp

**Data Availability Statement:** All relevant data are within the paper and its Supporting Information files.

## Abstract

Abdominal spacers are useful for maintaining the distance between the target tumors and surrounding tissues, such as the gastrointestinal tract, in patients treated with carbon ion radiotherapy. Surgical intervention to remove the spacers is sometimes necessary because of abdominal infections triggered by long-term spacer placement or intestinal perforation. Therefore, spacers that do not require surgical removal and provide effective drainage against abdominal infections are urgently needed. This study aimed to develop a spacer that could be removed non-surgically and one that provides the therapeutic effect of drainage in patients who receive carbon ion radiotherapy for abdominal tumors. A novel fan-shaped spacer was constructed from a film drain that was folded along the trigger line. Simple withdrawal of the trigger line caused the film drain to fold and the holding lines to become free. We performed laparoscopy-assisted insertion with pneumoperitoneum and blind removal of the spacer fourteen times using a porcine model. Saline in the abdominal cavity was effectively aspirated using the spacer. Our novel fan-shaped spacer could be removed safely without surgery and was able to drain fluid effectively from the abdominal cavity.

## Introduction

For radiotherapy (RT), including carbon ion radiotherapy (C-ion RT) or for intensity-modulated radiation therapy (IMRT) of abdominal tumors, safety margins are needed around the clinical target tumor volume. The margin extension strongly depends on the applied radiation

**Funding:** This work was supported by Grants-in-Aid for Scientific Research from the Japan Society for the Promotion of Science (JSPS); grant number 18K07665 to TY. The work was also supported in part by Research Grant of the Princess Takamatsu Cancer Research Fund, Suzuken Memorial Foundation, and Pancreas Research Foundation of Japan.

**Competing interests:** The authors have declared that no competing interests exist.

technique, the availability of image guidance, and selection of a patient positioning protocol, inter- and intra-fractional target motion, and dose-exposure limits of neighboring organs at risk [1]. Delivery of high doses to the surrounding healthy organs by adding safety margins may cause severe late toxicities such as ulceration and perforation in the gastrointestinal tract. To prevent these undesirable side effects, providing a radical dose to the target tumor volume has been frequently limited. Further development of related medical devices is required worldwide.

C-ion RT has been used for the treatment of several malignant tumors, including head and neck cancer, soft tissue tumors, and prostate cancer [2–4]. C-ion RT shows very steep dose gradients at the margin of the irradiated volume; however, carbon ion beams may induce severe tissue damage if the space between the target tumors and surrounding tissues is too small. In fact, rectal bleeding has been reported after C-ion RT for prostate cancer, probably because the prostate and rectum are anatomically proximate [5].

The small and large intestine, including the rectum, are relatively sensitive to RT; therefore, balloons and Gore-Tex sheets have been surgically inserted into the abdominal cavity as spacers to displace the intestinal tract from the target tumor. Existing spacers are useful devices for preventing injury to the surrounding tissue caused by radiation therapy, including heavy-ion radiotherapy; nevertheless, invasive surgery is often needed to remove these foreign bodies after finishing radiation or to avoid adverse events such as abdominal infection [6]. To avoid surgery for removing the spacer, spacer gels with the absorptive properties have been used [1]. However, surgical intervention to remove spacers or to drain infectious fluid collections from the abdominal cavity is necessary for radiation-treated patients with abdominal infections caused by long-term spacer placement [7]. Spacers that do not need to be surgically removed and yet effectively drain the abdomen are urgently needed.

Therefore, the purpose of this study was to develop not only a spacer that did not require surgical removal for C-ion RT, but also one that functioned as a therapeutic drain in patients with abdominal infections. We created a fan-shaped spacer by fixing a folded-film drain with lines out of materials that are installable into the abdominal cavity and that are easily removable without invasive surgery. To demonstrate the usefulness of the spacer, we used an animal model to determine whether the spacer could effectively drain fluid collected in the abdominal cavity and then be removed safely.

## Materials and methods

### Materials of the spacer

The spacer was made out of a film drain that is a flat type of silicon tube (NIHON SEAL BOND, Japan), a main drain that is a cylindrical silicon tube (AS, ONE, Japan), and a trigger line that is a polytetrafluoroethylene tube (NICHIAS Corporation, Japan). None of these materials were denatured by C-ion RT.

### Animal experiment

A standard laparoscopic surgical setup (provided by Olympus Medical Systems Corp., Tokyo, Japan) was used to perform laparoscopic surgery on two female pigs at the Showa University Sannoudai Hospital medical training educational center. Two healthy pigs obtained from a three-way cross including the Landrace, Large White and Duroc came from the Fujii farm (Ishioka, Ibaraki, Japan). The pigs weighed 40 kg and each pig was used on a different day. We created a 3-cm laparotomy in the lower abdominal midline incision to insert the spacer during isoflurane anesthesia. Pneumoperitoneum was established at 10 mmHg, and two laparoscopic trocars were inserted at both sides of the lower abdominal region. Laparoscopic surgery was

performed in a head-down position. The spacer was inserted to make the space between the bladder and the small intestine. All procedures were performed under general anesthesia with a zoological analyst and sacrificed by 20 mEq KCL injection. This experiment was approved by the animal experiment committee of the Showa University Sannoudai Hospital medical training educational center (**approval number: 20181208 and 20190119**).

## Results

### Structure of the novel fan-shaped spacer made of folded-film drain and two kinds of lines

Our novel fan-shaped spacer was constructed from a film drain, a main drain, and a trigger line (Fig 1A). Dots were plotted on a film tube to indicate the holes with distances of 40, 1.5, 2, 2, 2, 2, 1.5, 2, 2.3, 2.2, 2, 1.5, 2, 3, 3, 2, 1.5, 2, 3.5, 3.5, 2, 1.5, 2, 4, 4, 2, 1.5, 2, 4.5, 4.5, 2, 1.5, 2, 5, 5, 2, 1.5, 2, 5.5, 5.5, 2, 1.5, 2, 6, 6, 2, 3, 1, 3, 2, 13, 2, 3, 1, 3, and 2 cm in-between. Holes of 3.5 mm were made in the center of the film tube at the marked plots. A 3.5 mm hole was made 1 cm away from one end of the main drain. The film tube was inserted through the main drain from the side of the hole that was made. The first hole of the film tube was matched with the main drain's hole, and the trigger line was inserted through the main drain. The trigger line was inserted into the third hole of the film tube, and every fifth hole of the film tube, creating loops on both the right and left sides of the trigger line (Fig 1B). The end of the trigger line was clipped to hold the loops.

A string was wrapped at the root of the trigger line, and both ends of the string were taken and threaded on a needle. One of the sides was picked and the needle was inserted through all the holes of the film tube on the trigger line side. One end of the string was taken and wrapped around the last hole once from the front and reinserted in the same direction as the needle passed through. The other end was taken and reinserted similarly from behind. Both ends of the string were tied. Another string was hung on the string at the root of the trigger line. Both ends of the string were taken, and the needle was threaded.

Similarly, it was inserted through all the holes on the film tube closest to the trigger line. As noted above, both ends of the string were inserted and the ends of the string were tied. Another string was hung on the root of the trigger line, both ends of the string were taken and a needle was threaded. The needle was inserted through the hole of the main drain. Then, the needle was inserted through the two lateral holes of the outermost loop of the film tube from the inside. From the outside, the needle was inserted through the remaining holes towards the root of the trigger line. The strings were reinserted and tied similarly. Another string was hung on the string where the outermost loop of the film tube was attached to the main drain. Both ends of the string were taken to thread the needle. The needle was inserted through the hole on the main drain, then the two lateral holes of the film tube from the inside, and then the remaining holes towards the root as with the other side. The unnecessary parts of the strings were cut. The trigger line was pulled so that the end would be inside the outer surface of the film tube (video). The size of the fan shape part was 10 × 10 cm, with a thickness of 10 mm. This was connected to main drain (Fig 1C).

### Removing the fan-shaped spacer

The blue holding lines passed through the folded-film drain and were fixed as loops with the trigger line to maintain the fan shape (Fig 1D, right panel). The loop fixing of the trigger line and holding lines is one of the specific invention points because the simple withdrawal of the trigger line can free the folded-film drain and the holding lines (Fig 1E). The free folded-film

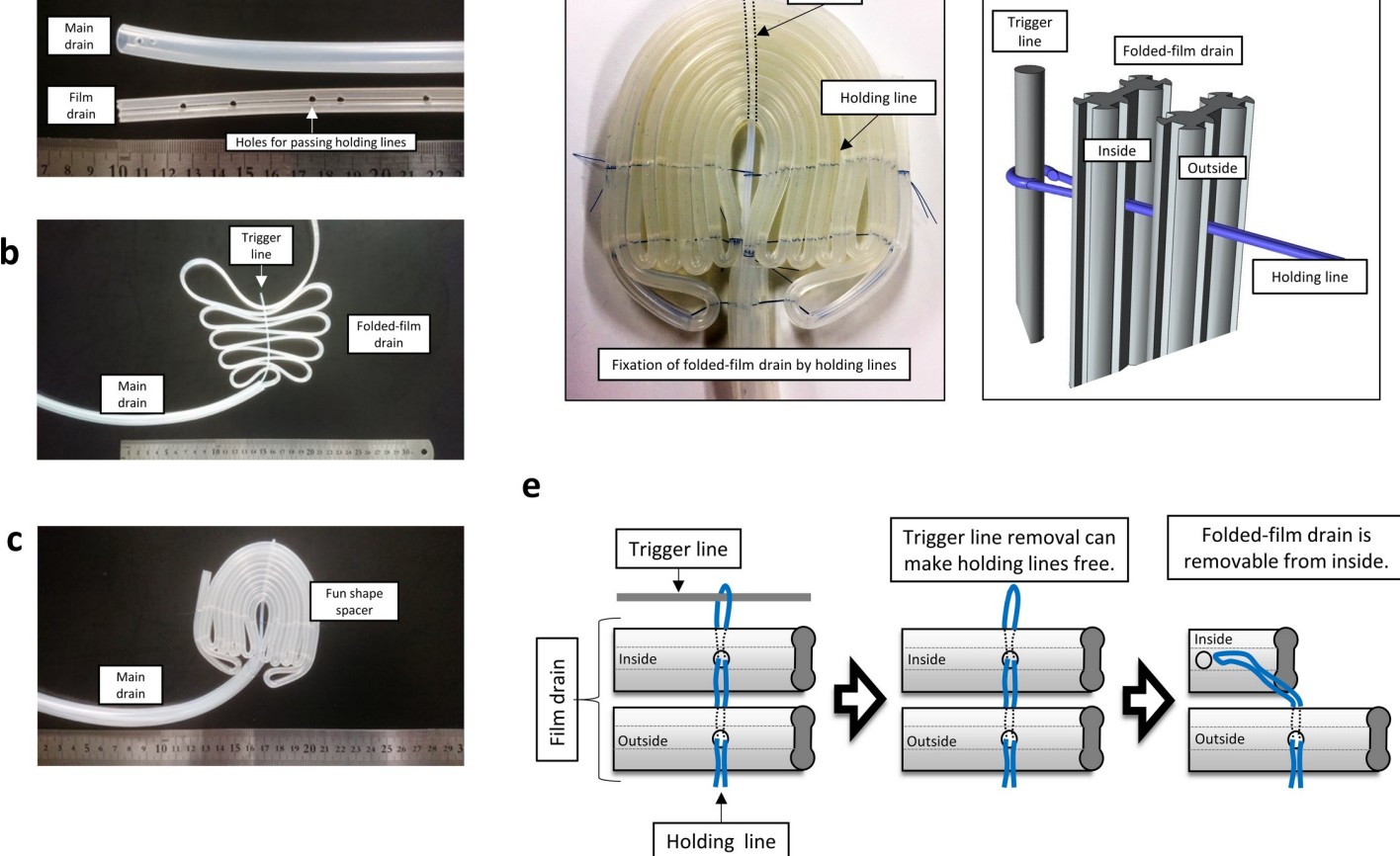

**Fig 1. Structure and removable mechanism of our novel fan-shaped spacer made of folded-film drain and two kinds of lines.** a) Materials for the fan-shaped abdominal spacer. b) Structure of the fan-shaped spacer before fixing a folded-film drain to a trigger line. These materials are installable into the abdominal cavity in the clinic. c) Fixed appearance of our novel spacer. d) Positional relationship of a folded-film drain, a trigger line, and holding lines in a fan-shaped spacer. Left panel: the folded-film drain is fixed to the trigger line by blue holding lines. Right panel: schema of the folded-film drain, the trigger line, and holding lines. e) Schema of the fan-shaped drain removal process without surgical procedure. The trigger line withdrawal out of the abdominal cavity makes the holding lines free, followed by the removal of folded-film drain from inside to outside.

drain was removable from inside-to-outside depending on the withdrawal of the main drain. The spacer form changed from fan-shaped to straight. The folded-film drain can be withdrawn with the main drain. The process for the sequential removal of the spacer from the pelvic cavity was demonstrated in a dummy model (Fig 2).

## Insertion and safe removal of the fan-shaped spacer in an animal model

We evaluated the usefulness of the fan-shaped spacer using a porcine model. The fan-shaped spacer was inserted in the abdominal cavity through a 3-cm laparotomy incision and was fixed to the abdominal wall of the pelvic cavity by suturing under laparoscopic assistance (Fig 3A). The small intestine was separated from the pelvic space. As shown in Fig 2, the withdrawal of the trigger line freed the folded-film drain (Fig 3B), and the free-film drain was removable from the abdominal cavity by simple withdrawal of the main drain (Fig 3C and 3D). To imitate the management of the abdominal drain in the clinic, we performed laparoscopy-assisted insertion of the spacer fourteen times using the porcine model with pneumoperitoneum and

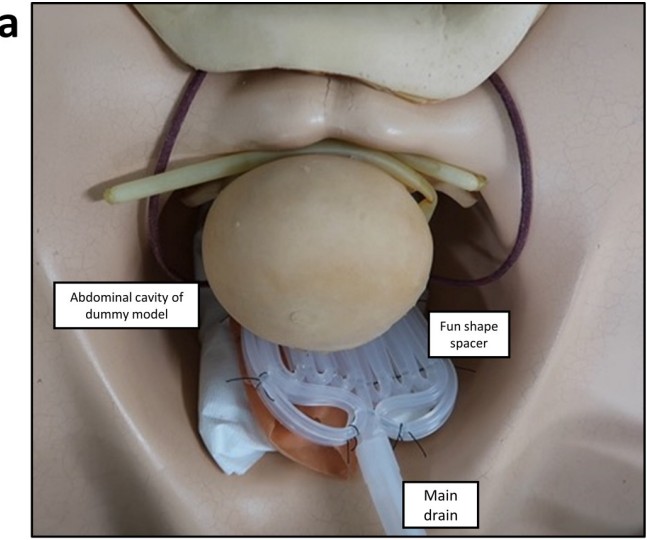

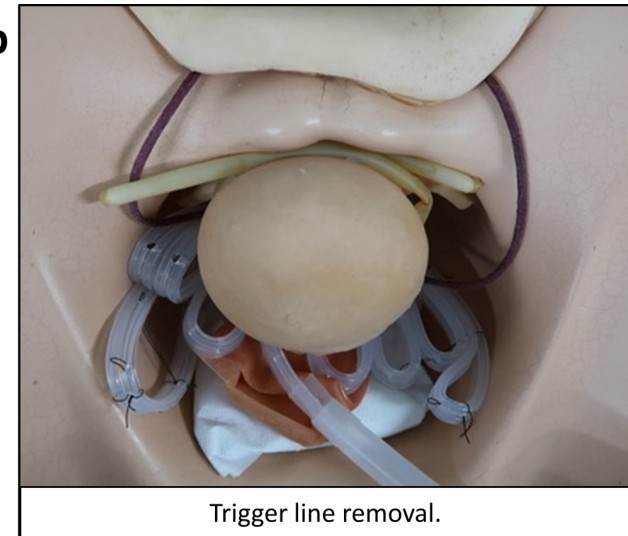

Trigger line removal.

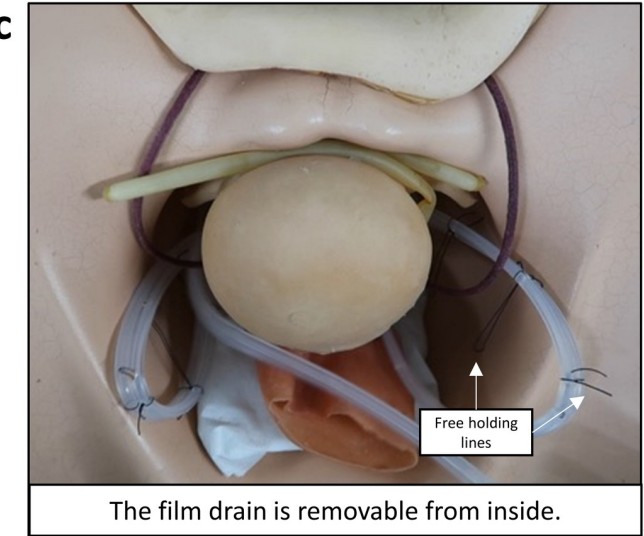

The film drain is removable from inside.

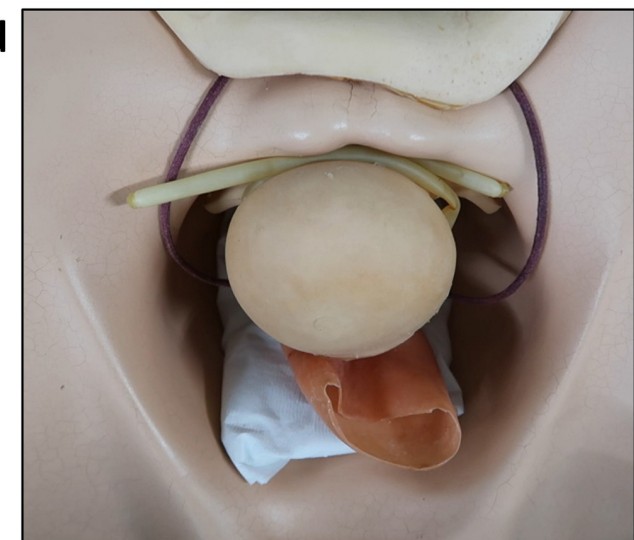

**Fig 2. Removal process of the fan-shaped spacer in the pelvic cavity of a dummy model.** a) Setup of the fan-shaped spacer in the pelvic cavity of a dummy model. b) Trigger line removal from the spacer can free the folded-film drain and the holding lines. c) The free film drain in the pelvic cavity is removable from the inside drain. d) After removal of the fan-shaped drain by simple withdrawal of the main drain.

blind removal without pneumoperitoneum. There was no obvious bleeding or abdominal organ injury caused by the insertion and removal of our spacer.

## Effective drainage of fluid collections in the abdominal cavity by the fan-shaped space

To evaluate the drainage efficacy of the spacer, we injected saline into the abdominal cavity of the porcine model (Fig 4A and 4B) and tried to aspirate the saline through the fan-shaped spacer. We were able to effectively aspirate saline from the abdominal cavity using the spacer (Fig 4C).

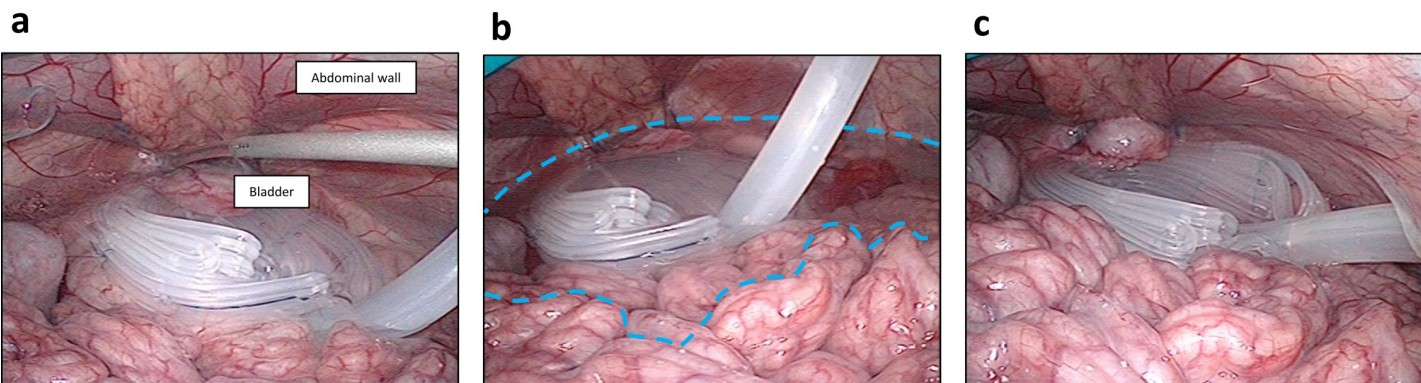

**Fig 3. Removal of the fan-shaped spacer in an animal model.** a) The fan-shaped spacer was inserted into the pelvic cavity of an animal model through a small 3-cm laparotomy. b) Trigger line removal from the spacer can free the folded-film drain and the holding lines. c) The free film drain in the pelvic cavity is removable from the inside drain. d) The fan-shaped spacer was removable from the abdominal cavity without bleeding or injury to surrounding organs by simple withdrawal of the main drain.

**Fig 4. Efficacy of fan-shaped spacer for the drainage of fluid collection in the abdominal cavity.** a) The fan-shaped spacer and saline representing collected fluid were set in the abdominal cavity of an animal model. b, c) The spacer was used to drain the collected fluid of the abdominal cavity in the animal model. Blue dot line indicates the surface level of the collected fluid.

## Discussion

In this study, we showed that our new fan-shaped spacer for C-ion RT and IMRT could be removed safely and easily without surgical intervention. Moreover, we showed that the spacer could drain the collected fluid in the abdominal cavity in the animal model. The spacer may be a new medical device that can be converted from a spacer to a therapeutic drain during the therapeutic course of radiation, depending on the patient's status and the status of the abdominal infection.

Long-term installation of spacers for radiation therapy causes several adverse events, including intestinal perforation and abdominal infection [6,7]. The spacer should be removed after finishing radiation therapy as soon as possible to prevent spacer-induced adverse events. Nevertheless, spacers have been installed for long periods after radiation treatment in many cancer patients because an invasive surgical procedure is necessary to remove existing spacers, including the balloon type and Gore-Tex sheets. To solve these problems, an absorbable spacer gel was developed; nevertheless, it is impossible to remove the spacer gel from the abdominal cavity if there is a change in patient status. In this study, we demonstrated the non-surgical removal of our fan-shaped spacer using a simple withdrawal of the main drain. These observations suggest that the fan-shaped spacer may prevent adverse events during therapy by virtue of this quick removal process at the end of radiation therapy.

Even if the patients are good candidates for radiation therapy with abdominal spacers, the abdominal installation of existing spacer types (mentioned above) is not suitable simultaneously with gastrointestinal anastomosis because the spacers act as foreign bodies that serve as foci of infection [6]. Therefore, if the spacers are needed for a post-operative therapeutic strategy, a second operation to install the spacer in the abdominal cavity must be performed in the clinic. For such cancer patients at high-risk of abdominal infection, our fan-shaped spacer may be installable without a second operation because the spacer can work as a therapeutic drain to remove infectious fluid caused by the intestinal anastomosis. In other words, our novel fan-shaped spacer may extend the indications for heavy ion beam therapy for cancer patients with high-risk of abdominal infection.

Several abdominal drains have been used in operations. The purpose of drain installation into the abdominal cavity is not only for treatment, but also to obtain information [8]. As mentioned above, our fan-shaped spacer may decrease the risk of abdominal infection during treatment because it is easily removable just after finishing radiation therapy without surgery. Nevertheless, if an abdominal infection occurs during radiation, the fan-shaped spacer may be used to diagnose the deep site infection rapidly. The other existing spacers do not have this function; therefore, our spacer has a substantial advantage for patients at high-risk of abdominal infection. Moreover, as shown in Fig 4, this spacer can drain infectious fluid from the abdominal cavity.

This study revealed the efficacy of our fan-shaped spacer for making distance between the target and surrounding organs, and our spacer could be removed without difficulty. Nevertheless, long-term indwelling experiment, securing the durability, and checking adverse events is necessary.

## Conclusions

In conclusion, we developed a novel fan-shaped spacer for heavy ion therapy and showed that the spacer could be removed safely without surgery; it can also drain fluid effectively from the abdominal cavity. By using this new medical device, the indications for heavy ion beam therapy and IMRT may be extended for cancer patients at high-risk of abdominal infections, ruling

out the use of existing balloons, Gore-Tex sheets, and absorbable spacer gels. Clinical trials to evaluate the safety and efficacy of the device in patients are urgently needed.

## Supporting information

**S1 Checklist. The ARRIVE guidelines checklist.**
(PDF)

**S1 Video.**
(WMV)

## Acknowledgments

We thank Ms. Mariko Nakamura and Mr. Yasuo Miyabe for their excellent assistance.

## Author Contributions

**Conceptualization:** Takayuki Asao.

**Data curation:** Navchaa Gombodorj.

**Project administration:** Ryo Takahashi, Hiroomi Ogawa.

**Supervision:** Naoya Ohta, Tatsuya Ohno, Hiroshi Saeki, Ken Shirabe.

**Writing – original draft:** Norio Kubo.

**Writing – review & editing:** Takehiko Yokobori.

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
