## [Decision Letter · Decision Letter 0]

17 Mar 2020

PONE-D-20-00215

An abdominal spacer that does not require surgical removal and allows drainage of infectious abdominal fluids in patients undergoing carbon ion radiotherapy

PLOS ONE

Dear Dr Kubo

Thank you for submitting your manuscript to PLOS ONE. After careful consideration, we feel that it has merit but does not fully meet PLOS ONE’s publication criteria as it currently stands. Therefore, we invite you to submit a revised version of the manuscript that addresses the points raised during the review process.

We would appreciate receiving your revised manuscript by 30 days. To enhance the reproducibility of your results, we recommend that if applicable you deposit your laboratory protocols in protocols.io, where a protocol can be assigned its own identifier (DOI) such that it can be cited independently in the future. For instructions see: http://journals.plos.org/plosone/s/submission-guidelines#loc-laboratory-protocols

We look forward to receiving your revised manuscript.

Kind regards,

Stefano Crippa

Academic Editor

PLOS ONE

Additional Editor Comments:

Please note that only two animals were used for the purpose of this study. Can you provide more experimental data?

The Editors also noted that authors make strong claims over the results by using language such as "epoch making device". Authors must rewording the relevant sections to reflect the need of further studies prior to validation of the presented device.

2. As part of your revision, please complete and submit a copy of the ARRIVE Guidelines checklist, a document that aims to improve experimental reporting and reproducibility of animal studies for purposes of post-publication data analysis and reproducibility:

https://www.nc3rs.org.uk/arrive-guidelines.

Please include your completed checklist as a Supporting Information file.

Please also include in the Methods the type of anaesthesia administered to animals.

Note that if your paper is accepted for publication, this checklist will be published as part of your article.

Reviewers' comments:

Reviewer's Responses to Questions

**Comments to the Author**

1. Is the manuscript technically sound, and do the data support the conclusions?

Reviewer #1: Yes

Reviewer #2: Partly

2. Has the statistical analysis been performed appropriately and rigorously? 

Reviewer #1: N/A

Reviewer #2: No

3. Have the authors made all data underlying the findings in their manuscript fully available?

Reviewer #1: Yes

Reviewer #2: Yes

4. Is the manuscript presented in an intelligible fashion and written in standard English?

Reviewer #1: Yes

Reviewer #2: Yes

5. Review Comments to the Author

Reviewer #1: The authors described the development of a home-made spacer to protect surrounding tissues/organs during radiotherapy, notably in the pelvis.

The paper is easy to read and clinically relevant.

I have some minor comments:

Please check the manuscript for spelling mistakes (some words are missing).

Remove "infectious" from the title.

Please add a video of how you make/construct the spacer. This point is very important to reproduce the technique

Reviewer #2: I read with interest the manuscript by Norio Kubo and collaborators about a new potential abdominal spacer to be used in patients undergoing radiotherapy. The authors show how this device acts both as a spacer, to perform local radiation of the tumor safely, as well as a drainage system to diagnose and treat eventual abdominal infections.

Additionally, the authors report that removal of this new fan-shaped spacer can be performed without surgery.

The authors should include some information about statistical methods they use and how they calculated their sample size. The study miss a control group, or alternatively, it could represent a control group for a further experiment.

6. PLOS authors have the option to publish the peer review history of their article (what does this mean?). If published, this will include your full peer review and any attached files.

Reviewer #1: Yes: Aurélien Dupré

Reviewer #2: No

---

## [Author Response · Author response to Decision Letter 0]

20 May 2020

Thank you for your constructive comments on our manuscript. The reviewers’ comments were very helpful. Our responses to the reviewers’ suggestions have been attached. We wish to express our appreciation to the reviewers for their insightful comments, which have helped us significantly improve our manuscript.

---

## [Decision Letter · Decision Letter 1]

28 May 2020

An abdominal spacer that does not require surgical removal and allows drainage of abdominal fluids in patients undergoing carbon ion radiotherapy

PONE-D-20-00215R1

Dear Dr. Kubo

We are pleased to inform you that your manuscript has been judged scientifically suitable for publication and will be formally accepted for publication once it complies with all outstanding technical requirements.

With kind regards,

Stefano Crippa

Academic Editor

PLOS ONE

Additional Editor Comments (optional):

Authors have addressed Reviewers' and Editor's comments and suggestions. The manuscript is suitable for publication.

Reviewers' comments:

Reviewer's Responses to Questions

**Comments to the Author**

1. If the authors have adequately addressed your comments raised in a previous round of review and you feel that this manuscript is now acceptable for publication, you may indicate that here to bypass the “Comments to the Author” section, enter your conflict of interest statement in the “Confidential to Editor” section, and submit your "Accept" recommendation.

Reviewer #1: All comments have been addressed

2. Is the manuscript technically sound, and do the data support the conclusions?

Reviewer #1: Yes

3. Has the statistical analysis been performed appropriately and rigorously? 

Reviewer #1: N/A

4. Have the authors made all data underlying the findings in their manuscript fully available?

Reviewer #1: Yes

5. Is the manuscript presented in an intelligible fashion and written in standard English?

Reviewer #1: Yes

6. Review Comments to the Author

Reviewer #1: The authors have responded the reviewers' requests.

The video is helpful to reproduce the technique which might be useful in surgical oncology.

7. PLOS authors have the option to publish the peer review history of their article (what does this mean?). If published, this will include your full peer review and any attached files.

Reviewer #1: Yes: Aurélien Dupré

---

## [Editor Report · Acceptance letter]

1 Jun 2020

PONE-D-20-00215R1 

An abdominal spacer that does not require surgical removal and allows drainage of abdominal fluids in patients undergoing carbon ion radiotherapy 

Dear Dr. Kubo:

I am pleased to inform you that your manuscript has been deemed suitable for publication in PLOS ONE. Congratulations! Your manuscript is now with our production department. 

With kind regards,

on behalf of

Dr. Stefano Crippa 

Academic Editor

PLOS ONE